# When And Where Are You Going?
# A Mixed-Reality Framework for Human Robot Collaboration

Shubham Sonawani
sdsonawa@asu.edu
Arizona State University
Tempe, Arizona, USA

Heni Ben Amor
hbenamor@asu.edu
Arizona State University
Tempe, Arizona, USA

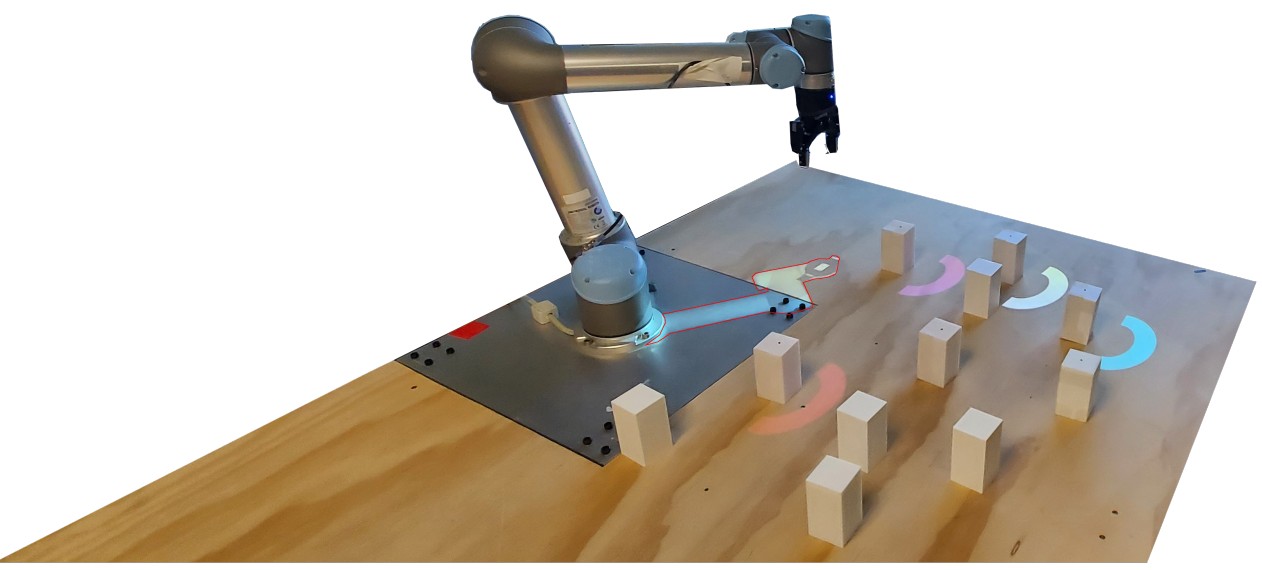

**Figure 1: Experiment Setup: Shadow Mode (outlined in red) and Highlight Mode (outlined in black) of Intention Projection framework**

## ABSTRACT

Fluency and coordination in human-robot collaborative tasks highly depend on shared situational awareness among the interaction partners. This paper sheds light on a work-in-progress framework for Intention Projection (IntPro). To this end, we propose a mixed-reality setup for Intention Projection that combines monocular computer vision with adaptive projection mapping to provide information about the robot's intentions and next actions. This information is projected in the form of visual cues into the environment. A human subject study consisting of a generic joint sorting task is proposed to validate the framework. Here, visual cues about the robot's intentions were provided to the human via mainly two modes, namely a) highlighting the object that the human needs to interact with

and b) visualizing the robot's upcoming movements. This work hypothesizes that combining these fundamental modes enables fast and effective signaling, which, in turn, improves task efficiency, transparency, and safety.

## KEYWORDS

Mixed-Reality, HRC, Human Subject Study

**ACM Reference Format:**
Shubham Sonawani and Heni Ben Amor. 2022. When And Where Are You Going? A Mixed-Reality Framework for Human Robot Collaboration. In . VAM-HRI 2022, (Virtual) Sapparo, Japapan, 4 pages.

## 1 INTRODUCTION

For humans and robots to effectively work together in close proximity, they need to have a mutual understanding of each other's intentions and actions. In traditional human-human interaction, the involved partners can learn to anticipate each other's actions through body language or timing. In human-robot teams, such an approach can lead to dangerous situations since robot movements are often hard to predict. Besides motion, other modalities can be used for signaling intent, e.g., visual or auditory modalities.

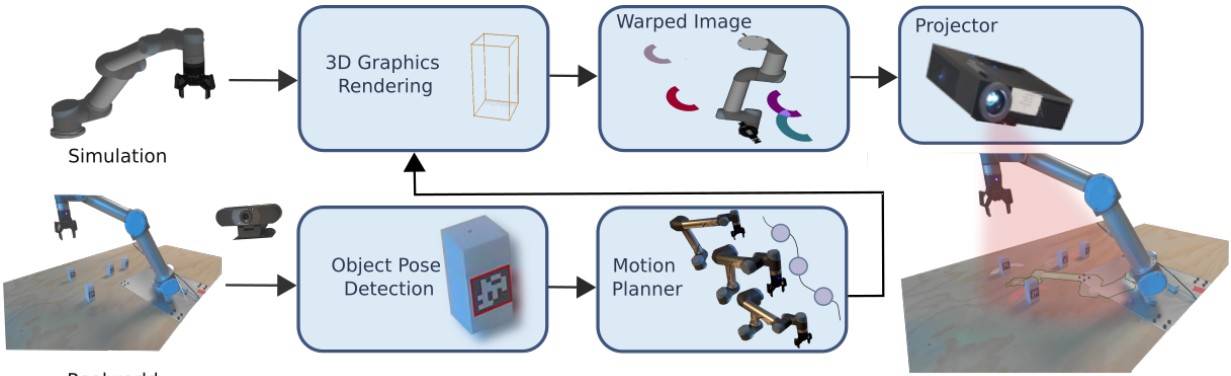

**Figure 2: Overview of Intention Projection Framework**

In this work, we focus on projecting visual cues via a mixed-reality approach. Building upon the concept of intention projection [1] we visualize the intention and future actions of the robot ahead of time. With regards to visual modalities, prior work focuses on discrete visual signals for providing robots intent. The work in [2] uses visual cues in the form of expressive lights on a Roomba robot to provide a complex interpretation of the robot. [3, 4] uses combinations of color and intensities of light signals to provide information about the robot's state given the current task and environment. To provide information in a continuous and feature-rich manner, projection mapping can be used to convey robot intent. [5] uses an onboard projector on a mobile robot to visualize navigation paths. The work in [6] provides a survey on the validity and informativeness of providing a robot's direction and velocity in the form of visual cues to the human during social navigation. [7] uses Nav-Points, Arrow, Gaze to communicate the robot's intention via head mounted display which showed a performance improvement in a human robot interaction tasks. Similarly [8] uses head-mounted mixed-reality setup to provide visualization of robot motion with respect to the user's frame of reference. This work compares 2D display versus head-mounted way mixed reality method backed up by user studies. Also, Situational awareness in proximal human-robot interaction was provided by [9] via augmented reality setup. Furthermore, prior work [10] uses object-aware projection technique to allow humans to collaborate effortlessly with the robot.

In this paper, we extend prior work on intention projection and investigate both discrete and continuous methods for communicating robot intent. In particular, we introduce a shadow mode in which a simulated robot (projected using mixed-reality) performs the intended robot actions ahead of the real, physical robot. In turn, the human user can visually anticipate the upcoming motion of the robot. We contrast this mode to a simple highlighting mode, in which the robot's *target object* is highlighted using a visual cue. We hypothesize that the visualization of a continuous motion (as performed in the shadow mode) provides a clearer and easier interpretation of the robot's intention when contrasted with a discrete highlighting of the target object. In this paper, we describe our setup for intention projection and the shadow mode and describe a planned human subject study to validate our hypothesis.

## 2 SYSTEM OVERVIEW

The overall system, as shown in Figure 2 consists of a combination of simulation and real-world setup for obtaining a final result of the IntPro framework. A monocular vision sensor combined with a structure light sensor such as a projector is used as a hardware setup to obtain information about the environment. Calibration sequence as per [11] is used to obtain an extrinsic matrix of projector with respect to camera and intrinsic parameters of the projector; for camera's intrinsic parameters, calibration is done separately to obtain more accurate parameters. This calibration is used by 3D-graphics rendering block to render real-world projection accurately. Simulation is used to render an image of the ur5 robot that mimics the same joint angles as the real-world ur5 robot. This rendered image is obtained from the bottom view of the simulated robot.

In order to dynamically change the rendering of highlighted objects, we leveraged an off-the-shelf pose detection framework [12], which can be expanded for multiple objects. Also, object pose information is fed to the motion planner, which calculates the inverse kinematics solution in the form of a trajectory of joint angles. These joint angles are provided simultaneously to the 3D graphics rendering block and real-world ur5 robot. Based on the delay set in shadow mode, the execution of the trajectory occurs on real-world and simulated ur5 robots. 3D graphics renderer keeps running and provides queried information based on the nature of the experiment.The set of experiments are designed considering the human subject and a combination of shadow and highlight mode. Detailed explanation about the proposed modes are given below:

- **Highlight Mode**: A 3D plane is rendered with the texture of a semi-elliptical disc and transformed onto the image frame with respect to the projector's frame of reference by using the detected object pose. Once projected onto the tabletop, the user can see the highlighted object in the form of a randomly colored elliptical disc in front of the object. This information can be provided simultaneously for all the objects with detected individual poses. Given real-time pose detection, highlight mode does not need objects to be static in the environment and projection can be adjusted to perturbation and change in the object's pose. Finally, This mode provides continuous information about the object of interest, which

keeps the user updated even when not looking at the object directly.

- **Shadow Mode**: Simulation rendering is leveraged to obtain the mirror effect of the robot onto the tabletop, which acts as the shadow of the robot. In order to provide shadow, the transformation between the tabletop and the camera frame is obtained by detecting the fiducial marker's pose. Since simulation and real world ur5 have the same joint angles, shadow mode dominantly shows the lateral trajectory of the robot, which helps the user understand which part of the work space is easy to engage in the task.

## 3 HUMAN SUBJECT STUDY DESIGN

In order to validate the efficacy of the framework, a human subject study will be conducted with generic sorting tasks. A user and robot will collaborate in a sorting task where the user will sort lightweight objects while the robot will sort heavyweight objects. These objects are 3D printed cubes with 10% (lightweight) and 50% (heavyweight) infill densities. Prior to the task, the user will be instructed about rules and conditions about the task. In the first part of the experiment, A user will be given instructions about the lightweight cubes' pattern on the display screen. The motivation behind these patterns is to reduce the complexity of the sorting task while keeping ambiguity as the sorting task progresses. These patterns can be created as shown in Figure 5. Now, as the robot starts a movement, the timer count for the sorting task begins, and the user should start sorting the objects. Once the human and the robot sort all the objects, the timer count stops.

In the second part of the experiment, explicit information about the pattern in which lightweight cubes are placed is not available to the user. However, details about highlight and shadow mode are given to the user. In only highlight mode, as shown in Figure 3, the user sorts the objects based on information provided via explicit and discrete visual cues onto the tabletop via projector. With only shadow mode, as shown in Figure 4, a user is implicitly informed about possible simple sorting patterns using the display screen. Here, the robot's trajectory information via shadow helps the user decide the safe part of the workspace to engage in sorting tasks. Furthermore, a delay of low, high, and no duration will be introduced between the shadow and the actual robot's movement in different sets of experiments with multiple users. This delay is hypothesized to help the user understand future robot's actions and plan the sorting task accordingly to reduce overall task execution time. Lastly, the sorting experiment will be performed with highlight and shadow mode. It is anticipated that the execution time of the proposed task will reduce significantly with the combination of shadow and highlight mode.

## 4 FUTURE WORK

This work aims to identify novel interaction and communication mechanisms for human-robot collaboration. We aim to develop visual mechanisms and languages that allow robots to project information about their state and the task into the environment. As a result, the physical world becomes a canvas. For many tasks, such visual communication could lead to a faster, more efficient, and more apparent introspection of the robot's beliefs and intentions. We

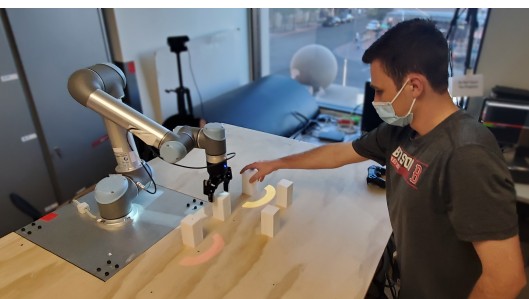

**Figure 3: Highlight Mode: A user collaborating with the robot while information about the next object explicitly provided via projections in the form of semi-elliptical disc.**

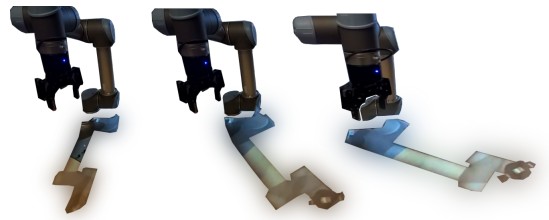

**Figure 4: Shadow Mode: Simulated version of the robot starts executing the intended motion before the physical robot. The simulated robot is projected into the table using our intention projection framework. This allows the human partner to preview the next actions and helps avoid collisions and increase transparency.**

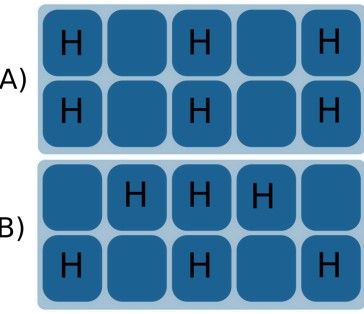

**Figure 5: Two different (A and B) patterns of 10 objects placement shown to the user prior to the experiment. Each square block in A) and B) signifies an object to be sorted, whereas block with the "H" mark needs to be sorted by the user and the rest by the robot**

will study how to best leverage such visual projections and which projection conventions are more fruitful than others. Our specific study mentioned in this paper focuses on the differences between *continuous*, animated projections versus discrete, static projections of information about the robot's next goals. We hypothesize that continuous projections lead to a faster and broader understanding of the current situations, whereas static projections may remain

ambiguous. This hypothesis will be carefully investigated in a user study as described above.

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
