# OpenReview forum: " When And Where Are You Going? A Mixed-Reality Framework for Human Robot Collaboration"
_humanrobotinteraction.org/HRI/2022/Workshop/VAM-HRI — VAM-HRI 2022_

### Official Review · Reviewer_8KAd · 2022-02-25
**Accept, good work**

**Rating:** 8
**Confidence:** 5

**Review:**

This paper is relevant to the VAM-HRI workshop and proposes a novel and interesting new method, and should be accepted. My suggestions for the authors are:

1. This paper should cite related work in MR for continuous visualizing modes such as visualizing robot trajectories [1], [2]. Also this work seems somewhat related to work in visualizing robot shadows in MR, such as [3].
2. In section 2, second paragraph, “Object pose” is incorrectly capitalized.
3. In Section 2 under highlight mode, sentence should be “this information can be provided simultaneous for all…” not to all.

[1] Walker, Michael, et al. "Communicating robot motion intent with augmented reality." Proceedings of the 2018 ACM/IEEE International Conference on Human-Robot Interaction. 2018.
[2] Rosen, Eric, et al. "Communicating and controlling robot arm motion intent through mixed-reality head-mounted displays." The International Journal of Robotics Research 38.12-13 (2019): 1513-1526.
[3] Boateng, Andrew, and Yu Zhang. "Virtual shadow rendering for maintaining situation awareness in proximal human-robot teaming." Companion of the 2021 ACM/IEEE International Conference on Human-Robot Interaction. 2021.

---

### Official Review · Reviewer_nuii · 2022-02-28
**Interesting concept builds on prior work**

**Rating:** 7
**Confidence:** 5

**Review:**

This paper briefly discusses a proposed human subjects study to evaluate the use of two different types of projection for understanding robot intent. The two types of projection are shadow, where a virtual shadow of the robot arm's movement is dynamically projected, and highlight, where the intended block to be grasped by the robot is highlighted with a static projection. Below are a few questions that will hopefully assist the authors with the details of their study.

1) What is the reasoning for your hypothesis? What elements of the related work lead you to believe the shadow mode will be preferable to the highlight mode?
2) Why is the color of the highlighting chosen randomly? Is there a rationale for this? Have you considered intentionally choosing the color(s)?

Minor edits:
- Please make sure to use the word "Figure" when referring to figure numbers throughout the paper.
- In the second sentence of Section 2, the phrase "project it back into it" is ambiguous. Please clarify what you mean here.

---

### Decision · Program_Chairs · 2022-03-04

Accept